# TRUSTED MULTI-VIEW CLASSIFICATION

**Zongbo Han, Changqing Zhang**,[*]
College of Intelligence and Computing
Tianjin University
Tianjin, China
`{zongbo,zhangchangqing}@tju.edu.cn`

**Huazhu Fu**
Inception Institute of Artificial Intelligence
Abu Dhabi, UAE
`hzfu@ieee.org`

**Joey Tianyi Zhou**
Institute of High Performance Computing
A*STAR, Singapore
`joey.tianyi.zhou@gmail.com`

## ABSTRACT

Multi-view classification (MVC) generally focuses on improving classification accuracy by using information from different views, typically integrating them into a unified comprehensive representation for downstream tasks. However, it is also crucial to dynamically assess the quality of a view for different samples in order to provide reliable uncertainty estimations, which indicate whether predictions can be trusted. To this end, we propose a novel multi-view classification method, termed trusted multi-view classification, which provides a new paradigm for multi-view learning by dynamically integrating different views at an evidence level. The algorithm jointly utilizes multiple views to promote both classification reliability and robustness by integrating evidence from each view. To achieve this, the Dirichlet distribution is used to model the distribution of the class probabilities, parameterized with evidence from different views and integrated with the Dempster-Shafer theory. The unified learning framework induces accurate uncertainty and accordingly endows the model with both reliability and robustness for out-of-distribution samples. Extensive experimental results validate the effectiveness of the proposed model in accuracy, reliability and robustness.

## 1 INTRODUCTION

Multi-view data, typically associated with multiple modalities or multiple types of features, often exists in real-world scenarios. State-of-the-art multi-view learning methods achieve tremendous success across a wide range of real-world applications. However, this success typically relies on complex models (Wang et al., 2015a; Tian et al., 2019; Bachman et al., 2019; Zhang et al., 2019; Hassani & Khasahmadi, 2020), which tend to integrate multi-view information with deep neural networks. Although these models can provide accurate classification results, they are usually vulnerable to yield unreliable predictions, particularly when presented with views that are not well-represented (*e.g.*, information from abnormal sensors). Consequently, their deployment in safety-critical applications (*e.g.*, computer-aided diagnosis or autonomous driving) is limited. This has inspired us to introduce a new paradigm for multi-view classification to produce trusted decisions.

For multi-view learning, traditional algorithms generally assume an equal value for different views or assign/learn a fixed weight for each view. The underlying assumption is that the qualities or importance of these views are basically stable for all samples. In practice, the quality of a view often varies for different samples which the designed models should be aware of for adaption. For example, in multi-modal medical diagnosis (Perrin et al., 2009; Sui et al., 2018), a magnetic resonance (MR) image may be sufficient for one subject, while a positron emission tomography (PET) image may be required for another. Therefore, the decision should be well explained according to multi-view inputs. Typically, we not only need to know the classification result, but should also be able to answer

---

[*]Corresponding author: Changqing Zhang

"How confident is the decision?" and "Why is the confidence so high/low for the decision?". To this end, the model should provide in accurate uncertainty for the prediction of each sample, and even individual view of each sample.

Uncertainty-based algorithms can be roughly divided into two main categories, *i.e.*, Bayesian and non-Bayesian approaches. Traditional Bayesian approaches estimate uncertainty by inferring a posterior distribution over the parameters (MacKay, 1992a; Bernardo & Smith, 2009; Neal, 2012). A variety of Bayesian methods have been developed, including Laplace approximation (MacKay, 1992b), Markov Chain Monte Carlo (MCMC) (Neal, 2012) and variational techniques (Graves, 2011; Ranganath et al., 2014; Blundell et al., 2015). However, compared with ordinary neural networks, due to the doubling of model parameters and difficulty in convergence, these methods are computationally expensive. Recent algorithm (Gal & Ghahramani, 2016) estimates the uncertainty by introducing dropout (Srivastava et al., 2014) in the testing phase, thereby reducing the computational cost. Several non-Bayesian algorithms have been proposed, including deep ensemble (Lakshminarayanan et al., 2017), evidential deep learning (Sensoy et al., 2018) and deterministic uncertainty estimate (van Amersfoort et al., 2020). Unfortunately, all of these methods focus on estimating the uncertainty on single-view data, despite the fact that fusing multiple views through uncertainty can improve performance and reliability.

In this paper, we propose a new multi-view classification algorithm aiming to elegantly integrate multi-view information for trusted decision making (shown in Fig. 1(a)). Our model combines different views at an evidence level instead of feature or output level as done previously, which produces a stable and reasonable uncertainty estimation and thus promotes both classification reliability and robustness. The Dirichlet distribution is used to model the distribution of the class probabilities, parameterized with evidence from different views and integrated with the Dempster-Shafer theory. In summary, the specific contributions of this paper are:

(1) We propose a novel multi-view classification model aiming to provide trusted and interpretable (according to the uncertainty of each view) decisions in an effective and efficient way (without any additional computations and neural network changes), which introduces a new paradigm in multi-view classification.

(2) The proposed model is a unified framework for promising sample-adaptive multi-view integration, which integrates multi-view information at an evidence level with the Dempster-Shafer theory in an optimizable (learnable) way.

(3) The uncertainty of each view is accurately estimated, enabling our model to improve classification reliability and robustness.

(4) We conduct extensive experiments which validate the superior accuracy, robustness, and reliability of our model thanks to the promising uncertainty estimation and multi-view integration strategy.

## 2 RELATED WORK

**Uncertainty-based Learning.** Deep neural networks have achieved great success in various tasks. However since most deep models are essentially deterministic functions, the uncertainty of the model cannot be obtained. Bayesian neural networks (BNNs) (Denker & LeCun, 1991; MacKay, 1992b; Neal, 2012) endow deep models with uncertainty by replacing the deterministic weight parameters with distributions. Since BNNs struggle in performing inference and usually come with prohibitive computational costs, a more scalable and practical approach, MC-dropout (Gal & Ghahramani, 2016), was proposed. In this model, the inference is completed by performing dropout sampling from the weight during training and testing. Ensemble based methods (Lakshminarayanan et al., 2017) train and integrate multiple deep networks and also achieve promising performance. Instead of indirectly modeling uncertainty through network weights, the algorithm (Sensoy et al., 2018) introduces the subjective logic theory to directly model uncertainty without ensemble or Monte Carlo sampling. Building upon RBF networks, the distance between test samples and prototypes can be used as the agency for deterministic uncertainty (van Amersfoort et al., 2020). Benefiting from the learned weights of different tasks with homoscedastic uncertainty learning, (Kendall et al., 2018) achieves impressive performance in multi-task learning.

**Multi-View Learning.** Learning on data with multiple views has proven effective in a variety of tasks. CCA-based multi-view models (Hotelling, 1992; Akaho, 2006; Wang, 2007; Andrew et al.,

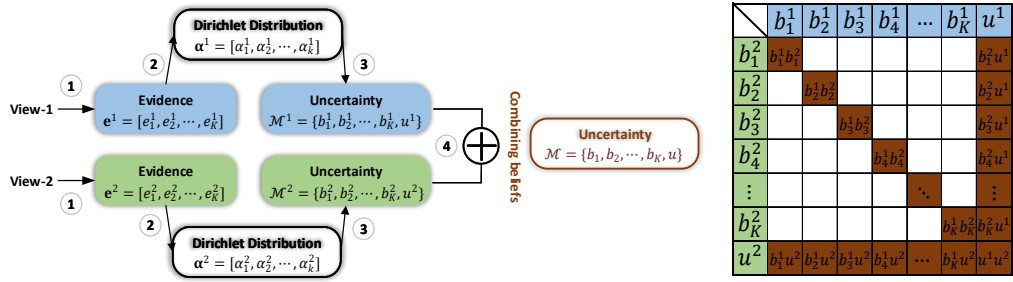

(a) Overview of the trusted multi-view classification  (b) Combining beliefs

Figure 1: Illustration of our algorithm. (a) The evidence of each view is obtained using neural networks (①). The obtained evidence parameterizes the Dirichlet distribution (②) to induce the classification probability and uncertainty (③). The overall uncertainty and classification probability are inferred by combining the beliefs of multiple views based on the DST (④). The combination rule and an example are shown in Definition 4 and (b), respectively. Given two sets of beliefs (blue and green blocks), we recombine the compatible parts of the two sets (brown blocks) and ignore the mutually exclusive parts (white blocks) of the two sets to obtain the combined beliefs.

2013; Wang et al., 2015a; 2016) are representative ones that have been widely used in multi-view representation learning. These models essentially seek a common representation by maximizing the correlation between different views. Considering common and exclusive information, hierarchical multi-modal metric learning (HM3L) (Zhang et al., 2017) explicitly learns shared multi-view and view-specific metrics, while $AE^2$-Nets (Zhang et al., 2019) implicitly learn a complete (view-specific and shared multi-view) representation for classification. Recently, the methods (Tian et al., 2019; Bachman et al., 2019; Chen et al., 2020; Hassani & Khasahmadi, 2020) based on contrastive learning have also achieved good performance. Due to its effectiveness, multi-view learning has been widely used in various applications (Kiela et al., 2018; Bian et al., 2017; Kiela et al., 2019; Wang et al., 2020).

**Dempster-Shafer Evidence Theory (DST).** DST, which is a theory on belief functions, was first proposed by Dempster (Dempster, 1967) and is a generalization of the Bayesian theory to subjective probabilities (Dempster, 1968). Later, it was developed into a general framework to model epistemic uncertainty (Shafer, 1976). In contrast to Bayesian neural networks, which indirectly obtain uncertainty through multiple stochastic samplings from weight parameters, DST directly models uncertainty. DST allows beliefs from different sources to be combined with various fusion operators to obtain a new belief that considers all available evidence (Sentz et al., 2002; Jøsang & Hankin, 2012). When faced with beliefs from different sources, Dempster's rule of combination tries to fuse their shared parts, and ignores conflicting beliefs through normalization factors. A more specific implementation will be discussed later.

## 3 TRUSTED MULTI-VIEW CLASSIFICATION

It has been shown that using a softmax output as confidence for predictions often leads to high confidence values, even for erroneous predictions since the largest softmax output is used for the final prediction (Moon et al., 2020; van Amersfoort et al., 2020). Therefore, we introduce an evidence-based uncertainty estimation technique which can provide more accurate uncertainty and allow us to flexibly integrate multiple views for trusted decision making.

### 3.1 UNCERTAINTY AND THE THEORY OF EVIDENCE

In this subsection, we elaborate on evidential deep learning to quantify the classification uncertainty for each of multiple views, which simultaneously models the probability of each class and overall uncertainty of the current prediction. In the context of multi-class classification, Subjective logic (SL) (Jøsang, 2018) associates the parameters of the Dirichlet distribution (Definition A.1 in the Appendix)

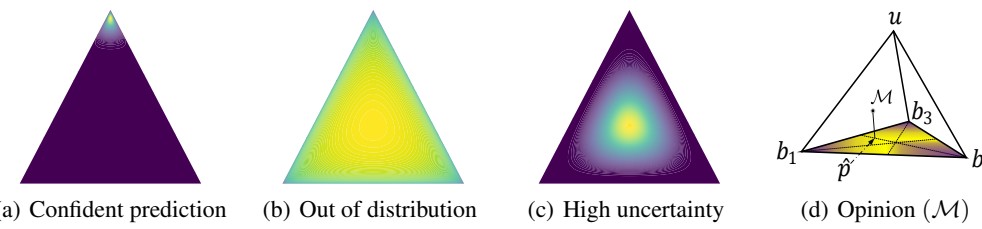

|                    |                       |                      |                  |
|--------------------|-----------------------|----------------------|------------------|
| (a) Confident prediction | (b) Out of distribution | (c) High uncertainty | (d) Opinion ($\mathcal{M}$) |

Figure 2: Typical examples of Dirichlet distribution. Refer to the text for details.

with the belief distribution, where the Dirichlet distribution can be considered as the conjugate prior of the categorical distribution (Bishop, 2006).

Accordingly, we need to determine the concentration parameters, which are closely related to the uncertainty. We elaborate on the Subjective logic (Jøsang, 2018), which defines a theoretical framework for obtaining the probabilities (belief masses) of different classes and overall uncertainty (uncertainty mass) of the multi-classification problem based on the *evidence* collected from data. Note that *evidence* refers to the metrics collected from the input to support the classification (step ① in Fig. 1(a)) and is closely related to the concentration parameters of Dirichlet distribution. Specifically, for the $K$ classification problems, subjective logic tries to assign a belief mass to each class label and an overall uncertainty mass to the whole frame based on the *evidence*. Accordingly, for the $v^{th}$ view, the $K + 1$ mass values are all non-negative and their sum is one:

$$u^v + \sum_{k=1}^{K} b_k^v = 1, \tag{1}$$

where $u^v \geq 0$ and $b_k^v \geq 0$ indicate the overall uncertainty and the probability for the $k^{th}$ class, respectively.

For the $v^{th}$ view, subjective logic connects the *evidence* $\mathbf{e}^v = [e_1^v, \cdots, e_K^v]$ to the parameters of the Dirichlet distribution $\boldsymbol{\alpha}^v = [\alpha_1^v, \cdots, \alpha_K^v]$ (step ② in Fig. 1(a)). Specifically, the parameter $\alpha_k^v$ of the Dirichlet distribution is induced from $e_k^v$, *i.e.*, $\alpha_k^v = e_k^v + 1$. Then, the belief mass $b_k^v$ and the uncertainty $u^v$ (step ③ in Fig. 1(a)) are computed as

$$b_k^v = \frac{e_k^v}{S^v} = \frac{\alpha_k^v - 1}{S^v} \quad \text{and} \quad u^v = \frac{K}{S^v}, \tag{2}$$

where $S^v = \sum_{i=1}^{K} (e_i^v + 1) = \sum_{i=1}^{K} \alpha_i^v$ is the Dirichlet strength. Eq. 2 actually describes the phenomenon where the more evidence observed for the $k^{th}$ category, the greater the probability assigned to the $k^{th}$ class. Correspondingly, the less total evidence observed, the greater the total uncertainty. The belief assignment can be considered as a subjective opinion. Given an opinion, the mean of the corresponding Dirichlet distribution $\hat{\mathbf{p}}^v$ for the class probability $\hat{p}_k^v$ is computed as $\hat{p}_k^v = \frac{\alpha_k^v}{S^v}$ (Frigyik et al., 2010).

**Differences from traditional deep-neural-network classifiers.** Firstly, the output of traditional neural network classifiers can be considered as a point on a simplex, while Dirichlet distribution parametrizes the density of each such probability assignment on a simplex. Therefore, with the Dirichlet distribution, SL models the second-order probability and uncertainty of the output. Secondly, the softmax function is widely used in the last layer of traditional neural network classifiers. However, using the softmax output as the confidence often leads to over-confidence. In our model, the introduced SL can avoid this problem by adding overall uncertainty mass. Existing methods (Gal & Ghahramani, 2016; Lakshminarayanan et al., 2017) usually require additional computations during inference to output uncertainty. Since the uncertainty is obtained during the inference stage, it is difficult to seamlessly train a model with high accuracy, robustness and reasonable uncertainty in a unified framework. Accordingly, the limitations underlying existing algorithms (*e.g.*, inability to directly obtain uncertainty) also limits their extension to trusted multi-view classification.

For clarity, we provide typical examples under a triple classification task to illustrate the above formulation. Let us assume that $\mathbf{e} = \langle 40, 1, 1 \rangle$ and accordingly we have $\boldsymbol{\alpha} = \langle 41, 2, 2 \rangle$. The

corresponding Dirichlet distribution, shown in Fig. 2(a), yields a sharp distribution centered on the top of the standard 2-simplex. This indicates that sufficient evidence has been observed to ensure accurate classification. In contrast, let us assume that we have the evidence $\mathbf{e} = \langle 0.0001, 0.0001, 0.0001 \rangle$, which is little evidence for classification. Accordingly, we obtain the Dirichlet distribution parameter $\boldsymbol{\alpha} = \langle 1.0001, 1.0001, 1.0001 \rangle$ and the uncertainty mass $u \approx 1$. As shown in Fig. 2(b), in this case, the evidence induces quite a flat distribution over the simplex. Finally, when $\mathbf{e} = \langle 5, 5, 5 \rangle$, there is also a high uncertainty, as shown in Fig. 2(c), even though the overall uncertainty is reduced compared to the second case. As shown in Fig. 2(d), we can convert a Dirichlet distribution into a standard 3-simplex (a regular tetrahedron with vertices (1,0,0,0), (0,1,0,0), (0,0,1,0) and (0,0,0,1) in $\mathbf{R}^4$) based on the subjective logic theory (Eq. 1 and Eq. 2), where the point ($\mathcal{M}$) in the simplex corresponding to $\left\{ \{b_k\}_{k=1}^3, u \right\}$ indicates an opinion. Accordingly, the expectation value $\hat{\mathbf{p}}$ of the Dirichlet distribution is the projection of $\mathcal{M}$ on the bottom.

## 3.2 DEMPSTER'S RULE OF COMBINATION FOR MULTI-VIEW CLASSIFICATION

Having introduced evidence and uncertainty for the single-view case, we now focus on their adaptation to data with multiple views. The Dempster–Shafer theory of evidence allows evidence from different sources to be combined arriving at a degree of belief (represented by a mathematical object called the belief function) that takes into account all the available evidence (see Definition 3.1). Specifically, we need to combine $V$ independent sets of probability mass assignments $\{\mathcal{M}^v\}_1^V$, where $\mathcal{M}^v = \left\{ \{b_k^v\}_{k=1}^K, u^v \right\}$, to obtain a joint mass $\mathcal{M} = \left\{ \{b_k\}_{k=1}^K, u \right\}$ (step ④ in Fig. 1(a)).

**Definition 3.1** *(**Dempster's combination rule for two independent sets of masses**) The combination (called the joint mass) $\mathcal{M} = \left\{ \{b_k\}_{k=1}^K, u \right\}$ is calculated from the two sets of masses $\mathcal{M}^1 = \left\{ \{b_k^1\}_{k=1}^K, u^1 \right\}$ and $\mathcal{M}^2 = \left\{ \{b_k^2\}_{k=1}^K, u^2 \right\}$ in the following manner:*

$$\mathcal{M} = \mathcal{M}^1 \oplus \mathcal{M}^2. \tag{3}$$

*The more specific calculation rule can be formulated as follows:*

$$b_k = \frac{1}{1-C}(b_k^1 b_k^2 + b_k^1 u^2 + b_k^2 u^1), u = \frac{1}{1-C} u^1 u^2, \tag{4}$$

*where $C = \sum_{i \neq j} b_i^1 b_j^2$ is a measure of the amount of conflict between the two mass sets (the white blocks in Fig. 1(b)), and the scale factor $\frac{1}{1-C}$ is used for normalization.*

The joint opinion $\mathcal{M}$ is formed based on the fusion of opinions $\mathcal{M}^1$ and $\mathcal{M}^2$. The joint belief mass of class $k$ ($b_k$) and overall uncertainty ($u$) correspond to the brown blocks in Fig. 1(b). Intuitively, the combination rule ensures: (1) when both views are of high uncertainty (large $u^1$ and $u^2$), the final prediction must be of low confidence (small $b_k$); (2) when both views are of low uncertainty (small $u^1$ and $u^2$), the final prediction may be of high confidence (large $b_k$); (3) when only one view is of low uncertainty (only $u^1$ or $u^2$ is large), the final prediction only depends on the confident view.

Then, given data with $V$ different views, we can obtain the above-mentioned mass for each view. Afterwards, we can combine the beliefs from different views with Dempster's rule of combination. Specifically, we fuse the belief mass and uncertainty mass between different views with the following rule:

$$\mathcal{M} = \mathcal{M}^1 \oplus \mathcal{M}^2 \oplus \cdots \mathcal{M}^V. \tag{5}$$

After obtaining the joint mass $\mathcal{M} = \left\{ \{b_k\}_{k=1}^K, u \right\}$, according to Eq. 2, the corresponding joint evidence from multiple views and the parameters of the Dirichlet distribution are induced as

$$S = \frac{K}{u}, e_k = b_k \times S \text{ and } \alpha_k = e_k + 1. \tag{6}$$

Based on the above combination rule, we can obtain the estimated multi-view joint evidence $\mathbf{e}$ and the corresponding parameters of joint Dirichlet distribution $\boldsymbol{\alpha}$ to produce the final probability of each class and the overall uncertainty.

**Advantages of using subjective logic compared with softmax.** Compared with softmax output, using subjective uncertainty is more suitable for the fusion of multiple decisions. Subjective logic provides an additional mass function ($u$) that allows the model distinguish between a lack of evidence. In our model, subjective logic provides the degree of overall uncertainty of each view, which is important for trusted classification and interepretability to some extent.

## 3.3 LEARNING TO FORM OPINIONS

In this section, we will discuss how to train neural networks to obtain evidence for each view, which can then be used to obtain the corresponding masses $\{\mathcal{M}^v\}_{v=1}^V$ and $\mathcal{M}$. The neural networks can capture the evidence from input to induce a classification opinion (Kiela et al., 2018), and the conventional neural-network-based classifier can be naturally transformed into the evidence-based classifier with minor changes. Specifically, the softmax layer of a conventional neural-network-based classifier is replaced with an activation function layer (*i.e.*, RELU) to ensure that the network outputs non-negative values, which are considered as the evidence vector $\boldsymbol{e}$. Accordingly, the parameters of the Dirichlet distribution can be obtained.

For conventional neural-network-based classifiers, the cross-entropy loss is usually employed:

$$\mathcal{L}_{ce} = -\sum_{j=1}^{K} y_{ij} \log(p_{ij}), \tag{7}$$

where $p_{ij}$ is the predicted probability of the $i$th sample for class $j$. For our model, given the evidence of the $i$th sample obtained through the evidence network, we can get the parameter $\boldsymbol{\alpha}_i$ (*i.e.*, $\boldsymbol{\alpha}_i^v = \mathbf{e}_i^i + 1$) of the Dirichlet distribution and form the multinomial opinions $D(\mathbf{p}_i|\boldsymbol{\alpha}_i)$, where $\mathbf{p}_i$ is the class assignment probabilities on a simplex. After a simple modification on Eq. 7, we have the adjusted cross-entropy loss:

$$\mathcal{L}_{ace}(\boldsymbol{\alpha}_i) = \int \left[ \sum_{j=1}^{K} -y_{ij} \log(p_{ij}) \right] \frac{1}{B(\boldsymbol{\alpha}_i)} \prod_{j=1}^{K} p_{ij}^{\alpha_{ij}-1} d\mathbf{p}_i = \sum_{j=1}^{K} y_{ij} \left( \psi(S_i) - \psi(\alpha_{ij}) \right), \tag{8}$$

where $\psi(\cdot)$ is the digamma function. Eq. 8 is the integral of the cross-entropy loss function on the simplex determined by $\boldsymbol{\alpha}_i$. The above loss function ensures that the correct label of each sample generates more evidence than other classes, however, it cannot guarantee that less evidence will be generated for incorrect labels. That is to say, in our model, we expect the evidence for incorrect labels to shrink to 0. To this end, the following KL divergence term is introduced:

$$KL\left[D\left(\mathbf{p}_i|\tilde{\boldsymbol{\alpha}}_i\right) \| D\left(\mathbf{p}_i|\mathbf{1}\right)\right]$$
$$= \log\left( \frac{\Gamma\left(\sum_{k=1}^{K} \tilde{\alpha}_{ik}\right)}{\Gamma(K) \prod_{k=1}^{K} \Gamma(\tilde{\alpha}_{ik})} \right) + \sum_{k=1}^{K} (\tilde{\alpha}_{ik} - 1) \left[ \psi(\tilde{\alpha}_{ik}) - \psi\left(\sum_{j=1}^{K} \tilde{\alpha}_{ij}\right) \right], \tag{9}$$

where $\tilde{\boldsymbol{\alpha}}_i = \mathbf{y}_i + (1 - \mathbf{y}_i) \odot \boldsymbol{\alpha}_i$ is the adjusted parameter of the Dirichlet distribution which can avoid penalizing the evidence of the groundtruth class to 0, and $\Gamma(\cdot)$ is the *gamma* function.

Therefore, given parameter $\boldsymbol{\alpha}_i$ of the Dirichlet distribution for each sample $i$, the sample-specific loss is

$$\mathcal{L}(\boldsymbol{\alpha}_i) = \mathcal{L}_{ace}(\boldsymbol{\alpha}_i) + \lambda_t KL\left[D\left(\mathbf{p}_i|\tilde{\boldsymbol{\alpha}}_i\right) \| D\left(\mathbf{p}_i|\mathbf{1}\right)\right], \tag{10}$$

where $\lambda_t > 0$ is the balance factor. In practice, we can gradually increase the value of $\lambda_t$ so as to prevent the network from paying too much attention to the KL divergence in the initial stage of training, which may result in a lack of good exploration of the parameter space and cause the network to output a flat uniform distribution.

To ensure that all views can simultaneously form reasonable opinions and thus improve the overall opinion, we use a multi-task strategy with following overall loss function:

$$\mathcal{L}_{overall} = \sum_{i=1}^{N} \left[ \mathcal{L}(\boldsymbol{\alpha}_i) + \sum_{v=1}^{V} \mathcal{L}(\boldsymbol{\alpha}_i^v) \right]. \tag{11}$$

The optimization process for the proposed model is summarized in Algorithm 1 (in the Appendix).

## 4 EXPERIMENTS

### 4.1 EXPERIMENTAL SETUP

In this section, we conduct experiments on six real-world datasets: Handwritten[1], CUB (Wah et al., 2011), Caltech101 (Fei-Fei et al., 2004), PIE[2], Scene15 (Fei-Fei & Perona, 2005) and HMDB

---

[1]https://archive.ics.uci.edu/ml/datasets/Multiple+Features
[2]http://www.cs.cmu.edu/afs/cs/project/PIE/MultiPie/Multi-Pie/Home.html

(Kuehne et al., 2011). We first compare our algorithm with single-view classifiers to validate the effectiveness of our algorithm in utilizing multiple views. Then, we apply existing classifiers to multi-view features and conduct experiments under different levels of noise to investigate their ability in identifying multi-view OOD samples. Details of these datasets and experimental setting can be found in the appendix.

**Compared methods**. We compare the proposed method with following models: (a) MCDO (monte carlo dropout) (Gal & Ghahramani, 2015) casts dropout network training as approximate inference in a Bayesian neural network; (b) DE (deep ensemble) (Lakshminarayanan et al., 2017) is a simple, non-Bayesian method which involves training multiple deep models; (c) UA (uncertainty-aware attention) (Heo et al., 2018) generates attention weights following a Gaussian distribution with a learned mean and variance, which allows heteroscedastic uncertainty to be captured and yields a more accurate calibration of prediction uncertainty; (d) EDL (evidential deep Learning) (Sensoy et al., 2018) designs a predictive distribution for classification by placing a Dirichlet distribution on the class probabilities.

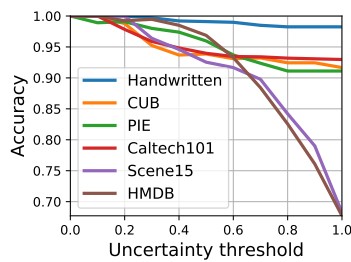

Figure 3: Accuracy with uncertainty thresholding.

## 4.2 Experimental Results

**Comparison with uncertainty-based algorithms using the best view.** We first compare our algorithm with current uncertainty-based classification methods. The detailed experimental results are shown in Table 1. Since most existing uncertainty-based classification methods use single-view data, we report the results of each method with the best-performing view in terms of both accuracy and AUROC (Hand & Till, 2001) to comprehensively compare our method with others. As shown in Table 1, our model outperforms other methods on all datasets. Taking the results on PIE and Scene15 as examples, our method improves the accuracy by about 7.6% and 14.8% compared to the second-best models (EDL/MCDO) in terms of accuracy respectively. Although our model is clearly more effective than single-view uncertainty-based models, it is natural to further ask - what happens if all algorithms utilize multiple views?

| Data | Metric | MCDO | DE | UA | EDL | Ours |
|------|--------|------|-----|-----|-----|------|
| Handwritten | ACC | 97.37±0.80 | 98.30±0.31 | 97.45±0.84 | 97.67±0.32 | **98.51±0.15** |
|  | AUROC | 99.70±0.07 | 99.79±0.05 | 99.67±0.10 | 99.83±0.02 | **99.97±0.00** |
| CUB | ACC | 89.78±0.52 | 90.19±0.51 | 89.75±1.43 | 89.50±1.17 | **91.00±0.42** |
|  | AUROC | **99.29±0.03** | 98.77±0.03 | 98.69±0.39 | 98.71±0.03 | 99.06±0.03 |
| PIE | ACC | 84.09±1.45 | 70.29±3.17 | 83.70±2.70 | 84.36±0.87 | **91.99±1.01** |
|  | AUROC | 98.90±0.31 | 95.71±0.88 | 98.06±0.56 | 98.74±0.17 | **99.69±0.05** |
| Caltech101 | ACC | 91.73±0.58 | 91.60±0.82 | 92.37±0.72 | 90.84±0.56 | **92.93±0.20** |
|  | AUROC | 99.91±0.01 | **99.94±0.01** | 99.85±0.05 | 99.74±0.03 | 99.90±0.01 |
| Scene15 | ACC | 52.96±1.17 | 39.12±0.80 | 41.20±1.34 | 46.41±0.55 | **67.74±0.36** |
|  | AUROC | 92.90±0.31 | 74.64±0.47 | 85.26±0.32 | 91.41±0.05 | **95.94±0.02** |
| HMDB | ACC | 52.92±1.28 | 57.93±1.02 | 53.32±1.39 | 59.88±1.19 | **65.26±0.76** |
|  | AUROC | 93.57±0.28 | 94.01±0.21 | 91.68±0.69 | 94.00±0.25 | **96.18±0.10** |

Table 1: Evaluation of the classification performance.

**Comparison with uncertainty-based algorithms using multiple views.** To further validate the effectiveness of our model in integrating different various views, we concatenate the original features of multiple views for all comparison methods. We add Gaussian noise with different levels of standard deviations ($\sigma$) to half of the views. The comparison results are shown in Fig. 4. As can be observed that when the data is free of noise, our method can achieve competitive results. After introducing noise to the data, the accuracy of all the comparison methods significantly decreases. Fortunately, benefiting from the uncertainty-based fusion, the proposed method is aware of the view-specific noise and thus achieves impressive results on all datasets. Therefore, the effectiveness for both clean and

noisy multi-view data is well validated. However, it will be more convincing to explicitly investigate the performance in uncertainty estimation.

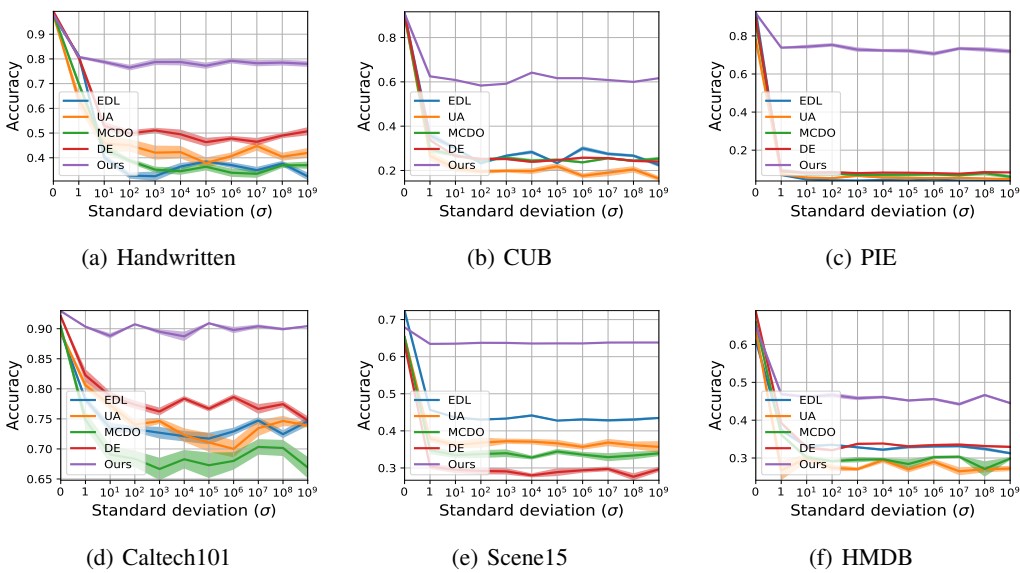

Figure 4: Performance comparison on multi-view data with different levels of noise.

**Uncertainty estimation.** To evaluate the uncertainty estimation, we visualize the distribution of in-/out-of-distribution samples in terms of uncertainty. We consider the original samples as in-distribution data, while the samples with Gaussian noise are viewed as out-of-distribution data. Specifically, we add Gaussian noise with the fixed level of standard deviations ($\sigma = 10$) to $50\%$ of the test samples. The experimental results are shown in Fig. 5. According to the results, the following observations are drawn: (1) Datasets with higher classification accuracy (*e.g.*, Handwritten) are usually associated with lower uncertainty for the in-distribution samples. (2) In contrast, datasets with lower accuracy are usually associated with higher uncertainty for the in-distribution samples. (3) Much higher uncertainties are usually estimated for out-of-distribution samples on all datasets. These observations imply the reasonability of our model in estimating uncertainty, since it can facilitate discrimination between these classes. Fig. 3 shows that our algorithm provides much more accurate predictions as the prediction uncertainty decreases. This implies that trusted decisions are supported based on the output (classification and its corresponding uncertainty) of our model.

## 5 CONCLUSION

In this work, we propose a novel trusted multi-view classification (TMC) algorithm which, based on the Dempster-Shafer evidence theory, can produce trusted classification decisions on multi-view data. Our algorithm focuses on decision-making by fusing the uncertainty of multiple views, which is essential for making trusted decisions. The TMC model can accurately identify the views which are risky for decision making, and exploits informative views in the final decision. Furthermore, our model can produce the uncertainty of a current decision while making the final classification, providing intepretability. The empirical results validate the effectiveness of the proposed algorithm in classification accuracy and out-of-distribution identification.

ACKNOWLEDGMENTS

This work was supported in part by National Natural Science Foundation of China (No. 61976151, No. 61732011), and the Natural Science Foundation of Tianjin of China (No. 19JCYBJC15200).

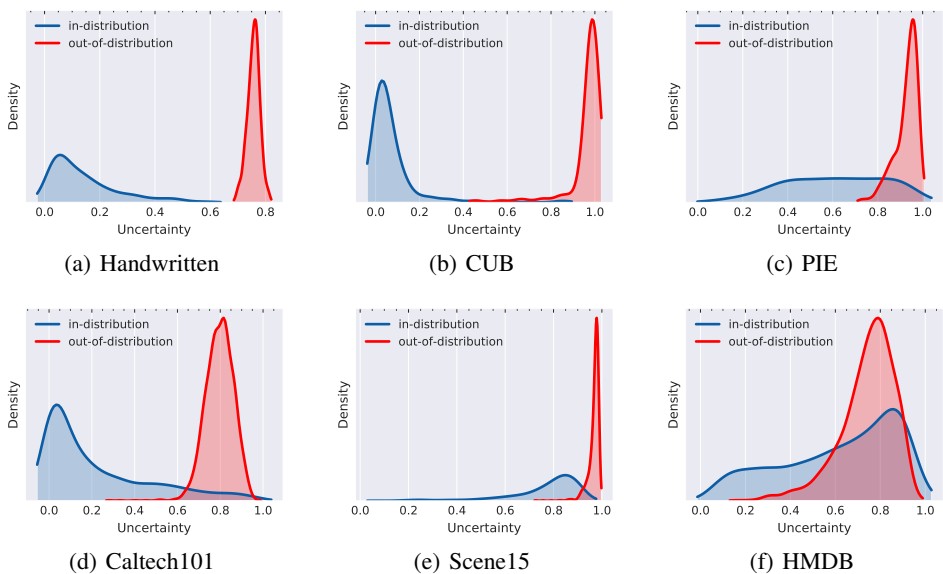

| (a) Handwritten | (b) CUB | (c) PIE |
| --- | --- | --- |
| (d) Caltech101 | (e) Scene15 | (f) HMDB |

Figure 5: Density of uncertainty.

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
