# OpenReview forum: "Trusted Multi-View Classification"
_ICLR.cc/2021/Conference — ICLR 2021 Poster_

### Official Review · AnonReviewer2 · 2020-10-27
**A reliable multi-view classification mechanism equipped with uncertainty**

**Rating:** 8
**Confidence:** 5

**Review:**

This paper proposes a reliable multi-view classification mechanism equipped with uncertainty, called Trusted Multi-View Classification. The goal is to dynamically assess the quality of different views for different samples to provide reliable uncertainty estimation. The idea is clear and well-motivated. The authors perform empirical studies on diverse datasets to conclude that the proposed algorithm is effective, robust and reliable.

Strengths:
+ It is interesting to conduct multi-view classification by dynamically integrating different views at an evidence level, which provides a novel and flexible way in multi-view classification.
+ The way of using Dempster-Shafer theory for integrating evidences in a unified and learnable framework is quite neat.
+ The paper is well-written and clearly presented.
+ Strong and sufficient empirical results are provided.

Minor comments:
+ It is reasonable to use the subjective logic theory to directly model uncertainty, however, beyond the advantages mentioned, it will be better if intuitive comparison or discussion between using u (overall uncertainty) and softmax scores in multi-view learning could be provided.
+ The authors could present failure cases which may be associated with high uncertainties (ideally).
+ For the results (Table 2) of the end-to-end experiments, are the data used original or being corrupted manually?


Overall, the paper is very well motivated and easy to follow. The assumptions and decisions are well supported. The stepwise experiments are helpful and provide good insights to evaluate the proposed algorithm. The method seems to be of great potential in real-world (cost-sensitive) applications.

---

> ### Author Response · Authors · 2020-11-15
> **Response to Reviewer 2**
>
> We appreciate for the identification of our novelty and the positive comments.
>
> Q1: It will be better if intuitive comparison or discussion between using u (overall uncertainty) and softmax scores in multi-view learning could be provided.
>
> R1: Thanks for the suggestion. More discussion and clarification of using u and softmax scores will be added. Specifically, the classification probability from the softmax function is usually too high for the misclassified samples [1]. When subjective uncertainty is introduced, the model will provide the degree of overall uncertainty, which is important for trusted classification.
>
> [1] Sensoy M, Kaplan L, Kandemir M. Evidential deep learning to quantify classification uncertainty[C]//Advances in Neural Information Processing Systems. 2018: 3179-3189.
>
> Q2: The authors could present failure cases which may be associated with high uncertainties (ideally).
>
> R2: In Table 3（Table 5 in the updated version）we provide several samples that associate with high uncertainty. We will follow the reviewers’ suggestion to add instructions on whether they are classified correctly in the end.
>
> Q3: For the results (Table 2) of the end-to-end experiments, are the data used original or being corrupted manually?
>
> R3: In the experiment in Table 2 (Table 4 in the updated version), we used the original data with noise. To avoid confusion, we have emphasized this in the revision.

---

### Official Review · AnonReviewer1 · 2020-10-28
**evidence level based multi-view classification**

**Rating:** 4
**Confidence:** 5

**Review:**

The authors propose using Dempster-Shafer evidence theory to build a more trusted multi-view classification method.

The article is fairly well structured, apart from the literature review, which somewhat does not discuss some of the related works properly, for example, the probabilistic CCA based methods and the optimal transport based papers [Dixin Luo, 2020].  The article has the unfortunate tendency to contains a few unproven (or wrong) claims. For instance, "[...] accurate classification results, they are usually vulnerable to yielding incorrect predictions, particularly when presented with views that are not well-represented [...]", which would deserve a citation or some empirical evidence. Specifically, I think some of the Bayesian versions of CCA methods are able to handle such a situation by providing uncertainty, and the authors did not include them as the baselines in the experimental results as well. It is also claimed that the proposed method is "optimal" sample-adaptive multi-view integration, which I could not see any proof in a theoretical manner.

My main concern with this paper is the experimental results. First of all, the paper is a kind of Bayesian multi-view learning method and should compare with Bayesian and deep CCA-based method as well. Second, the paper used a Dirichlet distribution for each view. Why HDP cannot be included as another baseline. Regarding the results, the authors emphasize that the performance has been improved by around 15%. This is not surprising as those methods are just single-view based methods.

Concerns/questions/comments:

1- The authors added Gaussian noise to half of the views. It is not clear to me that some views are noisy and some of them are not noisy. All of the features in one view are noisy, or all of the features are noisy.

2- Dirichlet distribution is somehow more suitable for the situation that one sample can belong to multiple classes. Did the authors consider that?

3- The integration methods, such as those proposed based on CCA and optimal transport (OT), try to combine different views somehow to achieve the best performance. However, the proposed method seems to investigate each view separately and then combine them. My concern is that when the results are noisy in all views, how the method performs. Is it helpful to combine them in this way? Adding some empirical experimental results would be great.

4- It is not clear to me that if the authors assume different views are well aligned or not. If not, can the proposed method handle such a situation?

5- The experiments if Figure 4 aim to show that the proposed method is more robust to the noisy environment. However, adding $\sigma = 10^9$ does not mean anything, and I think the method's performance is because it is not considering the view with the noise. But in real practice, when we talk about multi-view, both views are reasonably noisy. I prefer to see such a comparison. At least, a comparison in that way is also needed.

6- A comparison of the methods in semi-supervised classification would be great.

7- How the results change if you remove one of the views from a multiple-view (more than 2 views) dataset.

8- Last but not least, Table 1 seems to be a comparison with one-view based methods. This is not enough to show the proposed methods are good enough.


----------- UPDATE I ----------

I thank the authors for their response to my concerns/comments. It seems like I have to defend my position for suggesting a rejection of the paper. While the response of the authors has clarified some aspects, some comments have not been adequately addressed.

Still, the experiments and related works are my main concern with this paper.

R1: Based on this response, It seems the paper is trying to tackle the negative transferability problem. While this is an important problem, the authors need to discuss that in the related works, for example [1]. However, in a multi-view learning problem, the main issue is that each view includes some features which can help the classification goal. In other words, part of the features in all views are noisy. At least, you need to add this experiment in my point of view.

R10: [2,3,4] are a few examples that used DP for classification purposes. When one of the main arguments of the paper is adding uncertainty estimation using DP, discussing other related papers are important. In some cases, you can also compare with them.  Besides that, the authors can compare with HDP similar to what they did for the CCA-based methods.

It would be great if the authors could report the classification results based on only using one view for Figure 4 as a baseline. The authors might also want to check the paper [5].

[1] Bayesian multi-domain learning for cancer subtype discovery, NeurIPS 2018.

[2] Multi-Task Learning for Classification with Dirichlet Process Priors, Journal of Machine Learning Research 2007.

[3] Dirichlet-based Gaussian Processes for Large-scale Calibrated Classification, NeurIPS 2018.

[4] Factorial Multi-Task Learning : A Bayesian Nonparametric Approach, ICML 2013.

[5] Hierarchical Optimal Transport for Robust Multi-View Learning, NeurIPS 2020.

----------- UPDATE II ----------
I have read this paper again and went through the author's responses. While I appreciate the authors for their responses and believe there is some novelty in including uncertainty quantification, I'm still not confident with their experimental designs and literature review.

Based on my understanding, this method is not a multi-view learning in a classical way. It is better to say that the proposed method is somehow an ensemble approach. Going through the model, the paper deals with each view separately and then tries to combine the classification results of each view by weighting them. There is some novelty here, where the weighting is adapting based on the uncertainty. However, in a classical multi-view, the problem is that you have different views, where features of each view are randomly noisy. So, the goal is how one can combine information from different views to improve the classification results. Still, I cannot see that.

In the experiments, the authors make N views to be noisy, and the other N(+1) views are clean. Then they are trying to classify each view (somehow) separately and because they are reducing the weight of noisy views, their performance (slightly) will be improved. For example, please check table 1. For instance, the Handwritten dataset, the performance of DE, the only ensemble model as the baselines, performed almost similar 99.79 vs 99.97 ( 98.30 vs 98.51). This improvement can be due to tuning the other methods, and the authors did discuss this in neither the main text nor the supplement.

In Figure 4, when the noise is small, the different models' performances are almost the same. Also, \sigma = 10^9 is not meaningful and with some sort of variance comparisons, one can find that that view is corrupted.

The authors did not address if the method can work in real/harder situations when all features and views are noisy. The authors responded they do not have any restriction on that; however, they did not show either empirically or theoretically that the proposed method can handle this situation more suitable. I agree that the authors have tried on the real dataset, however, to see the performance comparison, you somehow need to randomly corrupt different features on different views.

Since the method does seem to be a multi-view ensemble learning, I would have rather expected to try a more common approach for achieving the same goal first, for example to use one of the available Bayesian single-view methods for each view and they combine the results and see how the results look like.

The authors claim that the negative transfer effect is not related here, which I can't entirely agree with. This term has been used in multi-task learning as well. The reason that no one uses this term in multi-view learning is that this is not a real scenario in multi-view learning. One view is degrading the classification performance because some views are noisy. If we do not use those very noisy views, the performance should be improved.

I also want to point out that their authors show that removing each view can degrade their model's performance. And this is desire, especially when the number of views is small, as having three views means three models in their ensemble architecture. It is not clear that this improvement is due to better information sharing or more complex methods.

Regarding the CCA-based method, one can use (Optimal) Bayesian Classification on learned space to get the uncertainty as well. Although, I agree that the proposed method is somehow an end-to-end learning method.

I would suggest the authors somehow re-organize/write the paper as a multi-view ensemble learning method and compare it with those multi-view ensemble methods as the main focus of the paper. You also need to discuss [5] as it is very related to your work. Although, it is a Bayesian but not an ensemble method.

---

> ### Author Response · Authors · 2020-11-15
> **Response to Reviewer 1 (2/2)**
>
> The concerns we sorted out in the comments:
>
> Q9: Concerns related to Bayesian CCA.
>
> R9: The CCA-based methods are quite different from ours. (1) Ours is a classification model while CCA-based methods are unsupervised representation learning models. (2) To the best of our knowledge, existing CCA-based methods are unable to provide trusted decisions. Therefore, we can compare ours with Bayesian and deep CCA-based methods only in terms of accuracy and we will do it and provide related discussion.
>
> Q10: Why HDP cannot be included as another baseline.
>
> R10: Hierarchical Dirichlet process (HDP) is a nonparametric Bayesian approach to clustering grouped data instead of classification method. Specifically, the HDP is quite different from the uncertainty-based classification tasks. (1) HDP is usually used for clustering while our method is used for multi-view classification. (2) HDP is usually used in grouped data, while our model is used for multi-view data. Therefore, it is difficult to conduct a fair comparison in experiments.
>
> Q11: Some concerns about writing.
>
> R11: (1) Empirical evidence about our claim "[...] accurate classification results, they are usually vulnerable to yielding incorrect predictions, particularly when presented with views that are not well-represented [...]": As shown in Figure 4, when some views are not displayed well, the prediction results are likely to be wrong. (2) Misunderstandings caused by the use of the word “optimal”: The meaning of “optimal” here means that our model can be optimized and ideally will obtain an optimal integration solution. To avoid confusion, we will change “optimal” to “promising” or “novel”. Thanks for the reviewer's suggestion.

---

> ### Author Response · Authors · 2020-11-15
> **Response to Reviewer 1 (1/2)**
>
> We appreciate the detailed comments. We believe the following point-to-point response can address all the concerns:
>
> Q1: The authors added Gaussian noise to half of the views. It is not clear to me that some views are noisy and some of them are not noisy. All of the features in one view are noisy, or all of the features are noisy.
>
> R1: In our experiments, there are datasets with two or more views. For a 2N-view dataset, all features in N views are corrupted with noise, while for (2N+1)-view datasets, N views are corrupted. We also conducted experiments on the naturally noisy data in the supplement (end-to-end experiment), where some of the samples are associated with noise on their all views.
>
> Q2: Dirichlet distribution is somehow more suitable for the situation that one sample can belong to multiple classes. Did the authors consider that?
>
> R2: We agree with the reviewer that the Dirichlet distribution may be applicable for the multi-label task which mainly focuses on how to effectively explore the correlation among different labels. In this work, we focus on the *multi-class* classification task, where each instance has only one label. The multi-label problem is a good direction for future work, but it out of the scope of the current work.
>
> Q3: The integration methods, such as those proposed based on CCA and optimal transport (OT), try to combine different views somehow to achieve the best performance. However, the proposed method seems to investigate each view separately and then combine them. My concern is that when the results are noisy in all views, how the method performs. Is it helpful to combine them in this way? Adding some empirical experimental results would be great.
>
> R3: When all views are noisy, traditional methods (e.g., CCA-based methods) may fail, while fortunately, it is one main advantage of our model for this case due to incorporating each view at evidence level. As shown in Fig. 5, our method can provide promising uncertainty estimation and thus can dynamically balance these noisy views. We also provide empirical results in Table 3 (Table 5 in the updated version). It is observed that when two views are all (heavily) noisy, our method produces high uncertainties for these samples. This is quite helpful for trusted classification.
>
> Q4: It is not clear to me that if the authors assume different views are well aligned or not. If not, can the proposed method handle such a situation?
>
> R4: In our work, different views are aligned which is the same as the setting in most existing multi-view learning methods. Our model does not pay attention to the task of view alignment which is another topic and out of the scope of this work.
>
> Q5: The experiments if Figure 4 aim to show that the proposed method is more robust to the noisy environment. However, adding \sigma=10^9 does not mean anything, and I think the method's performance is because it is not considering the view with the noise. But in real practice, when we talk about multi-view, both views are reasonably noisy. I prefer to see such a comparison. At least, a comparison in that way is also needed.
>
> R5: Fig. 4 provides the results when partial views are noisy, while note that the end-to-end experiments (in supplement) actually validate the effectiveness of our model in real applications. Specifically, Table 3 (Table 5 in the updated version) illustrates the examples that both views are noisy. It is observed that when two views are all (heavily) noisy, our method produces high uncertainties for these samples.
>
> Q6:  A comparison of the methods in semi-supervised classification would be great.
>
> R6: Our proposed method is a supervised classification algorithm, and currently we do not extend our method to semi-supervised learning. Thus, it is difficult to conduct a fair comparison with semi-supervised methods. Moreover, there are few multi-view semi-supervised classification models for estimating uncertainty.
>
> Q7: How the results change if you remove one of the views from a multiple-view (more than 2 views) dataset.
>
> R7: We will conduct experiments by removing a view manually.
>
> Q8: Table 1 seems to be a comparison with one-view based methods. This is not enough to show the proposed methods are good enough.
>
> R8: Table 1 is only a part of the experiment to show the advantage of multi-view learning. Actually, we also compared ours with methods using multiple views and the results are shown in Fig. 4. For clarification, we conducted experiments in a step-by-step manner in section 4.4: (1) Comparison with uncertainty-based algorithms using the best view (Table .1). (2) Comparison with uncertainty-based algorithms using multiple views (Fig. 4). (3) Uncertainty estimation (Fig. 3 and Fig. 5). Moreover, we also conducted experiments on real-world noisy datasets (see the end-to-end experiment in supplement).

---

> ### Author Response · Authors · 2020-11-25
> **Response to Reviewer 1 after update**
>
> We thank the reviewer for the update for the review. We now provide a point-to-point response to the left two concerns/comments as follows.
>
> Q1: “It seems the paper is trying to tackle the negative transferability problem…”
>
> R1: This is NOT a transfer learning or domain adaption model at all, therefore, we do not focus on tackling the negative transferability problem. Our work is a multi-view classification method that focuses on jointly using all views for trusted classification instead of transferring knowledge from one domain to another.
>
> There is no assumption about how many views (or how many features) are noisy. All views (no matter partial views or all views containing noise) can be adaptively integrated at the evidence level and this has been validated by sufficient experiments (including naturally noisy data in Table 4 and synthetic noisy data in Figure 4, 5).
>
> BTW: Negative transfer (NT), i.e., the source domain data/knowledge causes reduced learning performance in the target domain, which has been a long-standing and challenging problem in TL [1][2].
>
> [1] Overcoming Negative Transfer: A Survey, IEEE TKDE 2020
>
> [2] Catastrophic Forgetting Meets Negative Transfer: Batch Spectral Shrinkage for Safe Transfer Learning, NeurIPS 2019
>
>
>
> Q10: “[2,3,4] are a few examples that used DP for classification purposes. When one of the main arguments of the paper is adding uncertainty estimation using DP, discussing other related papers are important…”
>
> R10:  We would like to discuss the difference between the methods mentioned by the reviewer and ours.
> (1) We do not use DP (Dirichlet Process) at all in our model, we only use the Dirichlet distribution. So, there is no close relation between these models and ours.
> (2) All these models mentioned by the reviewer are single-view models and most of them are designed for other tasks (e.g., multi-task learning), so it is not proper to compare our method with these models.
> (3) We have compared ours with plenty of methods including the most related ones (e.g., EDL/UA/DE/MCDO and others).
>
> We will clarify these points in our revised version.
>
> Minor: “It would be great if the authors could report the classification results based on only using one view for Figure 4 as a baseline...”
>
> R：Fig. 4 is used to validate the ability to handle data with noisy view(s) under multi-view setting, so we do not add single-view baselines. To respond to the comment, we have conduct an experiment on two example datasets and the results are as follows:
>
> Handwritten
>
> |  method/noise ($\sigma$)   | 0  |  1 | 10 |
> |  ----  | ----  | ----  | ----  |
> | Ours  | 98.51±0.15 | 80.75±2.17 |78.75±3.35
> | EDL (multi-view)  | 98.00±0.13 |81.28±3.62 | 40.10±5.47
> | EDL (single-view)  | 97.67±0.32 | 16.46±1.81 | 13.14±0.57
>
> CUB
>
> | method/noise ($\sigma$)    | 0  |  1 | 10 |
> |  ----  | ----  | ----  | ----  |
> | Ours  | 91.00±0.42 | 62.50±1.27 |60.83±0.93
> | EDL (multi-view)  | 90.83±1.13 |35.83±7.06 | 30.00±2.44
> | EDL (single-view)  | 89.50±1.17 | 30.14±2.43 | 16.23±1.43
>
> It can be clearly observed that the results using multiple views are much better than those of using single view data.
> Due to time limitation, we will add baselines for the other datasets and update Fig. 4 in the final version.

---

### Official Review · AnonReviewer3 · 2020-10-29
**Interesting use of Dirichlet distribution for bringing uncertainty in output in multi-view multiclass classification**

**Rating:** 7
**Confidence:** 3

**Review:**

This paper has proposed a novel trust-based multi-view classifier. The idea of transferring classification output to the parameter of Dirichlet distribution to give uncertainty in output is novel and interesting. The model has interestingly used Dirichlet’s strength to define the weight of a view.

Paper is well written and mathematically sound. To my knowledge using Dirichlet distribution for bringing uncertainty in output in multi-view multiclass classification is new.

The proposed method has been also compared with recent existing works in the task of a single view and multiple view classification.

The proposed model has been motivated as classification with uncertainty in multi-view case. But in the Experiment section, it shows that even for single view setup the method is outperforming the existing models. It will be good to discuss the reason behind that. Why it is better than other methods in a single view case too?

Details of the experimental setup are important for the reproducibility of experimental results. It would have been better to have in the main paper.

---

> ### Author Response · Authors · 2020-11-15
> **Response to Reviewer 3**
>
> We appreciate for the identification of our contribution.
>
> Our model is a multi-view learning method, so through all this paper we use multiple views in our model. In Table 1, our model outperforms all compared ones that only using the best single view. Furthermore, in Figure 4, our algorithm is also competitive compared with other methods using multiple views especially under noisy cases. We also conducted experiments on the naturally noisy data in the supplement (end-to-end experiment).
>
> We will add more details of the experimental setup in the supplement, and release the code after acceptance.

---

### Public Comment · ~Xujiang_Zhao1 · 2020-11-13
**Question about Table 1**

Hi,

This paper is very well motivated and easy to follow. I have one question about Table 1, as Section 4.2 mentioned that "report the results of each method with the best-performing view ", does the proposed method also use the best-performing view? If yes, the proposed method (using only one view) would be the same as EDL, then how it outperforms EDL model in table 1?

---

> ### Author Response · Authors · 2020-11-15
> **Response to Question about Table 1**
>
> We appreciate for the identification of our novelty and the positive comments.
>
> Sorry for the confusion of Table. 1. Our model is a multi-view learning method, so through all this paper we use multiple views in our model. In Table 1, our model (using all views) outperforms all compared ones that only using the best single view.  Actually, we also compared ours with methods using multiple views and the results are shown in Fig. 4. For clarification, we conducted experiments in a step-by-step manner in section 4.4: (1) Comparison with uncertainty-based algorithms using the best view (Table .1). (2) Comparison with uncertainty-based algorithms using multiple views (Fig. 4). (3) Uncertainty estimation (Fig. 3 and Fig. 5). Moreover, we also conducted experiments on real-world noisy datasets (see the end-to-end experiment in supplement).
>
> We will clarify this in the revision. Thanks.

---

### Author Response · Authors · 2020-11-22
**Thanks and updated manuscript**

We would like to thank all the reviewers for their positive feedback on the novelty of our work and the significance in experiments.  We have updated the manuscript according to the reviewers’ suggestions. The new added experiments and implementation details are in the appendix.

---

### Decision · Program_Chairs · 2021-01-07
**Final Decision**

**Decision:**

Accept (Poster)

**Comment:**

The paper introduces a new idea for multi-view classification: using a Dirichlet distribution over the views to model uncertainty.

The paper appears to be clear, well written and sound.
Also, the experimental comparison is thorough.

The authors have given pertinent responses to the reviewers' questions, including w.r.t comparing against Bayesian/deep CCA in terms of accuracy.

Overall, this is a good paper.